# A practical evaluation of correlation filter-based object trackers with new features

**Islam Mohamed**[1,2], **Ibrahim Elhenawy**[1], **Ahmed W. Sallam**[2], **Andrew Gatt**[3], **Ahmad Salah**[1,4]*

**1** Faculty of Computers and Informatics, Zagazig University, Zagazig, Egypt, **2** Technical Research and Development Center, Cairo, Egypt, **3** Science Department, Eno River Academy, Hillsborough, NC, United States of America, **4** Department of Information Technology, CAS-Ibri, University of Technology and Applied Sciences, Muscat, Oman

* ahmad@hnu.edu.cn

**Data Availability Statement:** The relevant data for this study are available here: http://cvlab.hanyang.ac.kr/tracker_benchmark/datasets.html.

**Funding:** The author(s) received no specific funding for this work.

## Abstract

Visual object tracking is a critical problem in the field of computer vision. The visual object tracker methods can be divided into Correlation Filters (CF) and non-correlation filters trackers. The main advantage of CF-based trackers is that they have an accepted real-time tracking response. In this article, we will focus on CF-based trackers, due to their key role in online applications such as an Unmanned Aerial Vehicle (UAV), through two contributions. In the first contribution, we proposed a set of new video sequences to address two uncovered issues of the existing standard datasets. The first issue is to create two video sequence that is difficult to be tracked by a human being for the movement of the Amoeba under the microscope; these two proposed video sequences include a new feature that combined background clutter and occlusion features in a unique way; we called it *hard-to-follow-by-human*. The second issue is to increase the difficulty of the existing sequences by increasing the displacement of the tracked object. Then, we proposed a thorough, practical evaluation of eight CF-base trackers, with the top performance, on the existing sequence features such as out-of-view, background clutters, and fast motion. The evaluation utilized the well-known OTB-2013 dataset as well as the proposed video sequences. The overall assessment of the eight trackers on the standard evaluation metrics, e.g., precision and success rates, revealed that the Large Displacement Estimation of Similarity transformation (LDES) tracker is the best CF-based tracker among the trackers of comparison. On the contrary, with a deeper analysis, the results of the proposed video sequences show an average performance of the LDES tracker among the other trackers. The eight trackers failed to capture the moving objects in every frame of the proposed Amoeba movement video sequences while the same trackers managed to capture the object in almost every frame of the sequences of the standard dataset. These results outline the need to improve the CF-based object trackers to be able to process sequences with the proposed feature (i.e., *hard-to-follow-by-human*).

**Competing interests:** The authors have declared that no competing interests exist.

## 1 Introduction

Object tracking is a critical and difficult task. Thus, object tracking has become a popular topic of study in recent years. There are many uses for visual object tracking such as visual surveillance [1], video understanding [2], robotics [3], and unmanned aerial vehicle-based monitoring [4]. Object tracking has a significant impact in unmanned aerial vehicle missions, for example autonomous landing [5], tracking of flying vehicles [6], and target following [7], given a target in an axis-aligned [8–11] or rotated bounding box [12]. After the target bounding box has been defined in the beginning frame, the tracker must identify the patch that is the most similar in the next frames. The state is usually depicted as a bounding box enclosing the goal. Under adverse environmental conditions, most current trackers are still vulnerable. Occlusion, camera motion, clutter, similar appearance, and illumination effect are all problems faced by visual object trackers.

The tracking task is a mix of classification and estimation problems [13]. The first task is to use classification to provide a reliable location of the target. The following procedure is to calculate an accurate target location, which is frequently defined by the use of a bounding box. Visual object tracking techniques that are currently in use include discriminative correlation filters [14, 15], deep neural networks [16, 17], Siamese networks [18–20], and classic methods [21, 22], as shown in Fig 1. In this paper, we concentrate on CF-based trackers due to their computational efficiency and their real-time response. For instance, CF-based methods are the dominant object tracking technique in UAV tracking, which requires a fast tracker. Recently, the correlation filter has become one of the most common techniques in visual tracking for real-time applications. The correlation filter's effectiveness is primarily owing to its use of circulant matrices to perform potentially complex computation in the frequency domain instead of the spatial domain, to speed up the processing.

Recently, Correlation Filter Trackers (CFTs) have accomplished notable successes in terms of the reported accuracy rate on the standard datasets [23]. In the following study, we will provide a quick overview on some pertinent CFTs to illustrate this success. MOSSE tracker [24] is the oldest filter-based correlation method, which trains the filter with only grayscale samples. The CSK tracker [25] adds a kernel trick to the CF formula. With circulant shifted samples, the filter coefficients may be effectively optimized in the frequency domain. According to CSK [25], the KCF tracker [14] takes advantage of multi-channel HOG descriptors [26] to improve the representation of features capability as well as tracking efficiency. Likewise, in order to establish reliable tracking in color videos, color names (CN) features are added [27]. IBCCF

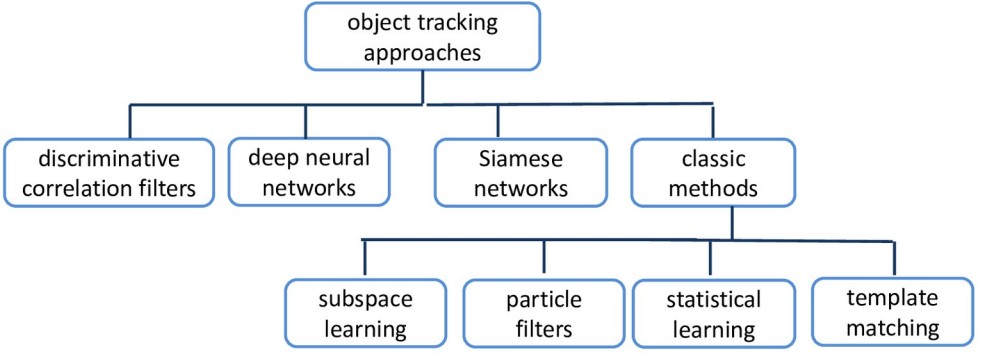

**Fig 1. A taxonomy of object tracking methods.**

[28], DSST [29], and SAMF [30] trackers use multi-scale searching techniques to deal with the scale adaptation issue. The authors in [31] proposed using dual filters, one for translation and another filter for scale variation.

The field of visual object tracking is a well-established field with several standard object tracking datasets (e.g., OTB-2013 [8], OTB-2015 [32], and VOT-2015 [33]) and successful CF-based trackers (e.g., MOSSE tracker [24]) and non-CF-based trackers [34]. The CF-based trackers are known for their low computational requirements and fast response time (i.e., locating the target in an image very fast). In this paper, we will focus only on CF-based visual trackers. The literature includes a very limited practical assessment of the CF-based visual tracker, especially for recent methods. Besides, The existing standard video sequences used for the practical evaluation the object trackers contain objects that are easy to be tracked by human beings. In addition, some of the standard video sequence datasets include objects that move with large displacement value; it is called fast motion feature. Video sequences with fast motion feature include a fast moving object but in the same direction. Thus, the performance of CF-based trackers should be evaluated when there is a large displacement in both object speed and direction. These are the major shortcoming of the CF-based object tracker which motivated the current work.

In this work, the need for exploring the performance of the CF-based object trackers on more difficult video sequences relative to the existing standard dataset motivated this work. In this context, we proposed two new video sequences for the movement of the Amoeba under the microscope. These two proposed video sequences have a new feature; they are difficult to track by human beings. We call this feature *hard-to-follow-by-human*, as the objects in the existing standard video sequences can be easily tracked by a human being. In addition, we proposed to extend the existing *fast motion* feature to increase the object displacement in different scenarios; we called this feature *very fast motion*. In addition, the lack of practical evaluation of the recent CF-based object trackers motivated this work as well. Thus, we provide a comprehensive practical evaluation of recent eight CF-based trackers, namely, LDES, BICF, CPCF, IBRI, MRCF, ARCF, DRCF, and TB_BICF. These eight trackers were evaluated using a variety of well-known benchmarks (i.e., OTB-2013) [8, 32] and the proposed video sequences as well. Then, the performance of each tracker was justified to outline their points of weakness and strength. Finally, we employed a success and precision plots [8]. The main contributions of this work can be summarized as follows:

1. We proposed a set of new video sequences with new attributes, namely, *hard-to-follow-by-human* and *very fast motion*. To our knowledge, it is the first time to propose evaluating object trackers with a video sequence which is hard to follow by human's eyes. The proposed video sequences and the results and trackers' source codes are is publicly available (https://github.com/elmesedi/A-practical-evaluation-of-correlation-filter-based-object-Trackers/).

2. We thoroughly practically evaluated eight recent CF-based trackers through a set of experiments, and the obtained results outlined the merits and demerits of these eight trackers.

The remainder of this article is structured as follows. The background is introduced in Section 2 and early work is reviewed. In Section 4, we will explained the evaluation metrics to evaluate the trackers. The dataset formulation is described in Section 3. Section 5 illustrates the assessment of the suggested trackers and the results of the experiment. Finally, we conclude the proposed work in Section 6.

## 2 Background and related work

### 2.1 Background

Many people are probably acquainted with the fundamentals of the correlation's principles, as they exist in statistics and probability. In pattern recognition, correlation is used to assess how similar or dissimilar a test target is to the training images. On the other hand, simple correlation is only effective when the test target and training images are similar. There is a demand for a wide range of approaches to enhance the fundamental correlation as well as obtain qualities for example flexibility with variances or distortions in reality (e.g., scale changes, image rotations, changes in light, etc.). The idea of correlation pattern recognition is explained in detail, as shown in Fig 2. There are two sub-images in Fig 2; a sub-image for testing with a variety of patterns, a test image *t*, and a template that we are searching for, a reference image *r*. In Fig 2, we search for the letter "S". Assume the two sub-images are binary, with white areas having a value of 0 and black regions having a value of 1. The sub-image for testing $t[x, y]$ and the template/reference image $r[x, y]$ are correlated as follows. We attempted aligning the template's top-left corner on the image's top-left corner, as shown in Fig 3.

Fig 3 includes one large image and one small sub-image. The small sub-image represents the patter, i.e., the object to be tracked, and the large image represents the image for testing, an image with many patterns. The small sub-image appears at the top-left corner of the large image; it is the image with kiwi fruit. The two images in Fig 3 are multiplied, pixel by pixel, and the product array's values are added to find the template's correlation value with the image for position of the two with relation to one another. Then, the values of correlation are calculated again by moving the template to all potential centerings with respect to the image. The correlation values are then determined again by moving the template to all feasible centerings with respect to the image. As a result, we can locate the desired targets by searching for peaks in the correlation output and determining if the peaks are large enough to detect the template's presence.

The simple correlation process is represented by Eq 1. We need to consider complicated correlation filters because the simple technique only works if the image contains perfect duplicates of the template image and no other items that have a similar appearance to the template. In Fig 2, for instance, the letter "O" and the letter "C" have a significant correlation, and for the letter "O" a simple correlation provides a high correlation result, which is inappropriate. Another drawback of the simple correlation process is that it can be extremely sensitive to

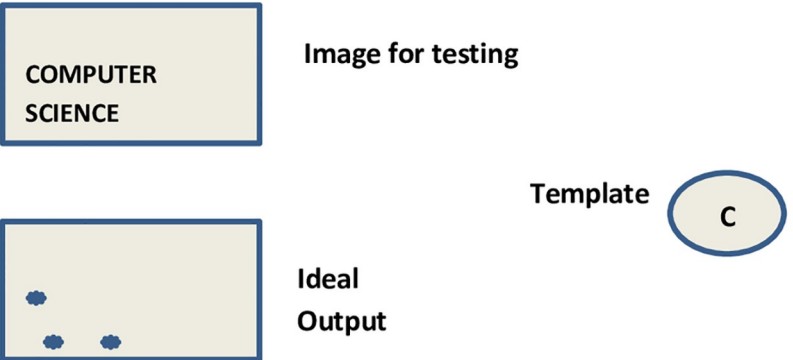

**Fig 2. The concept of image correlation.**

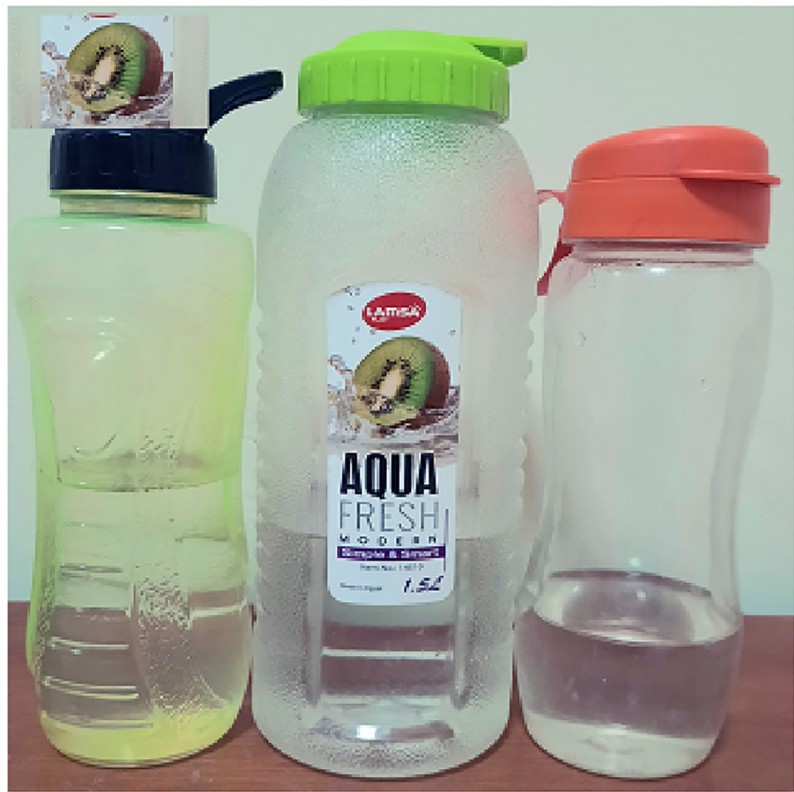

**Fig 3. Correlation pattern recognition.**

noise.

$$c[x, y] = \sum_i \sum_k t[k, 1] r[k + x, 1 + y] \tag{1}$$

Correlation outputs can lead to incorrect decisions if this randomness is not explicitly addressed. The simple correlation approach fails if the template finds massive modifications in appearance or shape in the target scene, often referred to as distortions, which could be caused by changes in lighting or viewing geometry (for example, scale changes, rotations, etc.). A good tracking system must be capable of dealing with such expected variation [35].

## 2.2 Related work

In 2010, for the first time, the correlation filter approach was employed in the object tracking. After nearly a decade of development, correlation filter tracking methods have reached maturity. Learning from gray-scale images can generate the minimum output sum of squared errors (MOSSE) [24]. For the first time, the correlation filter is applied to the tracking field. This filter is simple to compute and track the object rapidly, but it cannot predict accuracy when the appearance of the object changes.

In 2012, Henriques et al. introduced circulant structure tracking with kernels (CSK) [25]. MOSSE is a very reliable and efficient tracking system, and it has a low computational complexity that can process a large number of frames per second. In [36], the authors compared MOSSE and DCFs, where the performance of MOSSE was found to be better. On the other hand, MOSSE has several drawbacks; it cannot ensure that it will track an object accurately

when its appearance changes. In addition, MOSSE does not consider the target's historical data, instead, it updates the filter parameters with the previous and current frames only [37]. Thus, MOSSE tracker's robustness is low as well as its accuracy. Consequently, due to the accumulate error of MOSSE, the drifting problem increases which can not be reversed, and as a result the object will be lost; this is called anti-drift ability, which is low for the MOSSE method [37].

The MOSSE filter was improved by CSK [25] by introducing kernel approaches. The main limitation of CSK that it is limited to a single channel. Thus, the kernel correlation filter (KCF) [14] modified channel characteristics to multi-channel features and provides CN features for tracking. The CN feature [27] enhances the filter's capability to discriminate. In spite of the filter's ability to adapt to fast motion, out-of-view, and rotation, it still has to be enhanced. It is reported that KCF is sensitive to moving backgrounds, occlusion, and noise [38]. Of note, KCF utilized a constant size of the tracking bounding box. Its performance is limited when the tracked object size changes; in other words, KCF does not have the scale-invariant property [39].

To handle the multi-scale variation problem, Danelljan et al. [40] suggested a discriminative scale space tracker (DSST) based on the feature pyramid, as well as an enhanced method (fDSST) [29]. The authors proposed utilizing sub-grid interpolation of correlation values; this significantly reduced the method's processing time and the target search space with no effect on the proposed method's accuracy or robustness. One limitation of the DSST method is its poor performance to decide the target size when the object size changes frequently [41]. Regrading fDSST, the only improvement over DSST is the tracking speed [42].

By using phase correlation, Li et al. [43] estimated the target's scale and rotation by transforming the sample into log polar space. LDES [43] is a new tracker based on a correlation filter with a new method for estimating robust similarity transformations on massive displacements. LDES tracker divides the search space of the object into two sub-spaces. LDES performs well with a video sequence containing rigid objects, but its performance decays with occlusion [44].

The BICF method [45] is proposed to develop a new bidirectional incongruity-aware correlation filter. The bidirectional incongruity error, that is dependent on the response, is integrated into the correlation filter. BICF has the ability to quickly learn to adjust in looks, as well as suppression of inconsistent errors. Then, TB_BICF is proposed as a variant of BIFC in [46]. The TB_BICF proposed a novel bidirectional incongruity-aware correlation filter using the nature of tracking reversibility. Two drawbacks of the BICF and TB_BICF are reported in [46]. First, BICF could not track the object when it went out of view and reappeared again. Second, the BICF's performance is poor when the video sequence has an occlusion feature.

DRCF [47] proposes using a dual regularization method to dynamically regularize the filter that performs the correlation process within the core of the filter that produces ridge regression. DRCF proposed using a dual regulation strategy to reduce the regression correlation of the operation's border effect. DRCF utilizes a saliency-aware regularizer to handle any possible change in the size of the tracked object using an effective saliency detection technique. The main drawbacks of DRCF are discussed in [47]. First, rapid illumination effects the robustness of the DRCF tracker. Second, the fast scale change has a negative effect on the accuracy of the saliency detection.

IBRI tracker [48] was motivated by the fact that most of the existing methods combine the information of only two successive video frames; this approach limits the usefulness of the information, as using more frames increases the information and improves the object's boundary robustness. In addition, some methods utilized historical data from the previous video frames but without removing the noise of these frames. The authors of IBRI proposed a new

aerial object tracking method that takes advantage of disruptor-aware interval-based response inconsistency. Thus, IBRI tracker performance is good even when scale variation is presented.

CPCF [49] introduces a new method for pursuing dynamic consistency in correlation filters. CPCF utilizes HoG [26] and CN [27] features. CPCF utilized a consistency map to measure the level of consistency for each two successive video frames. This is preformed in a dynamic manner to extract rich temporal information.

MRCF [50] is a new tracking approach that has been proposed depending on a correlation filter with a multi-regularizers. The MRCF method is aware of the channel reliability. Thus, it can accomplish an automatic channel weight distribution through learning the distribution of each channel weight. It is mainly designed to be used as a UAV tracker.

To date, CPCF, MRCF, and IBRI trackers are not well evaluated, as they are recent methods. Thus, there are no reported drawbacks to date for this tracker. This motivated the present work to evaluate these methods alongside other trackers with good performance.

## 3 Dataset formulation

Object tracking benchmark datasets have been proposed to standardize the task of assessments of visual object tracking methods. They include video sequences with a variety of target categories, as well as varied time lengths and difficult attributes These datasets provide a wide range of video sequences, frames, attributes, and classes (or clusters). One of the most common benchmarks is the OTB-2013 dataset [8], which consists of 50 difficult video sequences labelled with 11 attributes. The sequences in the OTB-2013 datasets present several tracking challenges such as occlusion, deformation, and background clutters.

The summary of the proposed video sequences is listed in Table 1. First, modification of the existing video sequence was proposed by increasing the object displacement in consecutive frames. In other words, we proposed to modify the fast motion feature into the very fast motion feature. Thus, we selected two sequences from the OTB-2013 dataset already have the FM feature, namely, Tiger1, Surfer, and one sequence lacking the FM feature, i.e., Football1. This will help us to evaluate the performance of the trackers by increasing the motion speed for both the fast and slow motion objects.

In the following text, all the original sequence are from OTB-2103, unless otherwise is mentioned. The first proposed video sequence is based on Tiger1 sequence; it is called "Tiger1_VFM_1". The original Tiger1 sequence already has the FM property, and we proposed increasing the object displacement, speed, by removing the last ten frames out of every 50 frames. Thus, the proposed sequence includes a sudden change in the object's motion speed relative to the original Tiger1 sequence. Subsequently, the object's motion speed return to

**Table 1. The proposed video sequences.**

| Sequence Name | Sequence description | Attributes |
|---|---|---|
| Tiger1_VFM_1 | Drop the last ten frames from every 50 frames. | OPR, IV, DEF, MB, OCC, FM, VFM, IPR |
| Tiger1_VFM_2 | Drop ten frames when the object reach to the end of the view. | IV, MB, OCC, DEF, IPR, FM, VFM, OPR |
| Surfer_VFM | Drop the last 5 frames of every 20 frames, from the first 40 frames only. | SV, FM, VMF, IPR, OPR, LR |
| Football1_Modf | Drop the first frame from every 10 frames. | IPR, OPR, BC, FM |
| AMB1 | A clear object is occluded by a semi-opaque object. | OPR, BC, DEF, FM, |
| AMB2 | A blurred object is occluded by a semi-opaque object. | OPR, BC, DEF, FM |

normal in the next 40 frames and so on. We aim to explore whether the tracker can still detect a sudden change in the object's displacement.

In the second proposed sequence, the change will be observed in the motion speed as well as direction. The original Tiger1 sequence includes a predictable path of the tiger object. In the proposed sequence, a sudden change was posited in both the object's motion speed and direction by dropping ten frames when the tiger object reaches the end of the view. That results in reversing the direction of the object path. We call this proposed sequence "Tiger1_VFM_2". The third proposed sequence is similar to the first proposed sequence apart from two main differences. First, the original sequence is the Surfer. Second, rate of changing the motion speed is higher, as we proposed dropping the last five frames out of each 20 frames. We called this sequence "Surfer_VFM".

The fourth proposed sequence, and the last proposed modified sequence of the OTB-2103 dataset, is based on a sequence that does not have the FM feature. The aim of this selection is to evaluate the effect of increasing the speed of a slow moving object. The original sequence is the Football1; we proposed dropping the first frame from every 10 frames which is called "Football1_Modf".

The fifth and sixth proposed video sequences, in Table 1, are new sequences are not included in any existing dataset. The proposed sequences are extracted from a video capturing the Amoeba movement under a microscope. There is an intuitive impression that the sequences of the standard dataset can be easily tracked by a human beings. Thus, we aim to explore the performance of the CF-based trackers when the sequence has this property. In this context, we proposed extracting the two sequences from a video for life under the microscope of amoeba (https://www.youtube.com/watch?v=ZpAbk__xaew). The proposed two sequences based on this video are called AMB1 and AMB2, where AMB stands for Amoeba. AMB1 sequence consists of 20 frames from seconds 1:37 to 1:40, and AMB2 consists of 20 frames from seconds 1:25 to 1:27 of the aforementioned video. A sample frame of each of the two sequences are illustrated in Fig 4. The two sequences have the following features: out-of-plane rotation, deformation, and background clutters attributes, and occlusion.

The sequence AMB1, Fig 4a, is easier to detect than AMB2, Fig 4b, as the object in AMB1 is clear, while the object in AMB2 is blurred and the object's color resembles the background color. Occlusion is classified as inter-occlusion, background occlusion, and self-occlusion [51]. The proposed AMB1 and AMB2 push the limit of the classes of the occlusion feature. The intuition of proposing the AMB1 and AMB2 sequences is that the moving amoeba is entered into a larger object which should occult the amoeba, but as the object is not fully opaque, the amoeba object is not fully occult, and hard to be tracked by human's eyes. In other words, the proposed sequences, AMB1 and AMB2, combine two different features which are occlusion and background clutters. The merger of these two features results in a new feature called *semi-opaque occlusion*. In this feature, the occluding object can have a transparency level that makes it hard to track the object but not impossible for human's eyes. In other words, the occlusion

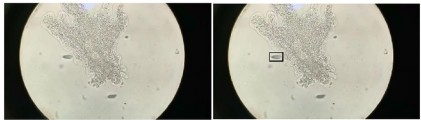
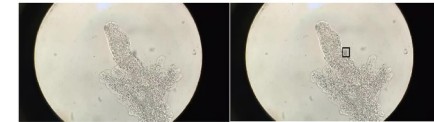

**(a)** The AMB1 sequence.                          **(b)** The AMB2 sequence.

**Fig 4. The proposed amoeba video sequences, clear image to the left and annotated image to the right.** a) AMB1 b) AMB2.

feature was binary; we proposed to make this feature with several levels of transparency, rather than restricting its identity to an occlusion or otherwise.

## 4 Evaluation metrics

We evaluated the trackers of interest using a familiar benchmark datasets [8]. The benchmark datasets contains 50 challenging image sequences with 11 attributes: out-of-view (OV), fast motion (FM), scale variation (SV), low resolution (LR), deformation (DEF), illumination variation (IV), background clutters (BC), motion blur (MB), occlusion (OCC), out-of-plane rotation (OPR), and in-plane rotation (IPR). Each attribute provides a unique challenge to object tracking. Many attributes can be assigned to one sequence. The benchmark OTB-2013 uses success rate and the precision for quantitative evaluation. Furthermore, it evaluates the robustness of object tracking in two ways.

### Precision plot

The center location error is a metric commonly used for assessing tracking precision. It calculates the difference between the tracker's predicted center $X_K^T$ and the ground-truth box center $X_K^G$ in the $k$-th frame. It is commonly referred to as RMSE (root mean square error):

$$RMSE = \sqrt{\frac{1}{N}\sum_{K=1}^{N}\|X_K^G - X_K^T\|} \tag{2}$$

where $N$ represents the total number of frames.

The average center location error across all frames of a particular sequence is then utilized to describe the overall efficiency of that sequence. The output location can be uncertain when the tracker misses the target, and the average error number might not always be useful to predict tracking efficiency [52]. The precision plot [25, 52] is a new way of evaluating global tracking performance. It displays the frames percentage in which the predicted position is located inside a specific range of the ground truth. As a representative precision score for each tracker, we choose the threshold score = 20 pixels [52].

### Success plot

The bounding box overlap another assessment measure, given ground truth box $b_a$, where the tracked box $b_t$ and the overlap score(S) is calculated as follows:

$$S = \frac{|b_t \bigcap b_a|}{|b_t \bigcup b_a|} \tag{3}$$

where $\bigcap$ and $\bigcup$ represent the intersection and union of two different parts, respectively, the region's pixel count is represented by |.|. We count how many successful frames have an overlap S that exceeds the defined threshold $t_o$ to assess a series of frames' performance. The success plot shows the ratios of successful frames at the thresholds varied from 0 to 1. For tracker evaluation, using a single value for the success rate at a certain threshold (for example, $t_o = 0.6$) could not be accurate or representative. Rather, we rate the tracking algorithms based on the area under the curve (AUC) Of every success plot.

### Robustness evaluation

The standard technique of assessing trackers is to execute them through a test sequence that begins with the ground truth location in the first frame and ends with producing the average

success rate or accuracy. This is called a one-pass evaluation (OPE). A tracker, however, may be sensitive to initialization and its efficiency might decrease or significantly increase when initializing differently at a different starting frame. Thus, OTB-2013 [8] recommends two methods for evaluating the robustness of a tracker to initialization, by temporally perturbing the initialization (starting at various frames) and spatially (starting by various bounding boxes). Spatial Robustness Evaluation (SRE) and Temporal Robustness Evaluation (TRE) are the terms for these tests. SRE and TRE are better than the traditional OPE approach, which only initializes every tracker on the initial frame; in TRE, each tracker is started on a separate frame (i.e., with a temporal spread); whereas a noisy bounding box in the SRE tracker is initialized.

## 5 Experiments and analysis

### 5.1 Setup

The proposed experiments were carried out on OTB-2013 [8] benchmark and our proposed video sequences in order to assess the benefits of the eight CF-based tracking algorithms. The experiments were performed on a desktop PC with Intel(R) Core(TM) i5–2410M CPU and RAM of 10GB using MATLAB 2017. In the proposed assessments of the CF-based trackers, we chose eight trackers with presently available source codes, namely, DRCF [47], BICF [45], TB_BICF [46], LDES [43], IBRI [48], MRCF [50], CPCF [49], and ARCF_H [53]. The utilized video sequences include a set of original five video sequences from the OTB-13 dataset, namely, Tiger1, Surfer1, Football1, DragonBaby, Ironman, and Box video sequences. In addition, we utilized four modified video sequences, namely, Football_Modf Tiger1_VFM_1, Tiger1_VFM_2, and Surfer_VFM. Besides, two new video sequences, namely, AMB1 and AMB2.

We used the resultant tracking information produced by OTB-2013 benchmark to compare the experimental results of the source codes of the eight CF trackers. Occlusion (OCC), motion blur (MB), background clutters (BC), fast motion (FM), and illumination variation were the features considered when comparing these trackers.

The results considering comparing the CF-trackers qualitative and quantitative analysis. The qualitative comparison includes frames with boundary boxes of the different trackers. The quantitative analysis includes three metrics, namely, 1) the precision and success plots of the OPE, TRE, and SRE, 2) the number of frame per second (fps), and 3) the number of frames where the tracker managed to capture the object. OPE is a standard assessment approach in which on each sequence is processed by the trackers only once. For TRE, it executes trackers on 20 sub-sequences separated by the parent sequence with various lengths, and SRE assesses trackers by starting with scaled ground or significantly modified truth bounding boxes in the first frame.

### 5.2 Discussion

First, for the qualitative results, Figs 5–7 show different performance of the eight trackers on different video sequences. Considering Figs 5 and 6 (http://cvlab.hanyang.ac.kr/tracker_benchmark/datasets.html), these two figures show that the proposed modified video sequences are more difficult for the tracker than the original video sequences. For instance, the performance of the trackers on OTB-2013 original sequence Tiger1 in Fig 5a is much better than the performance of the same trackers on the proposed modified sequence in Fig 5b and 5c. Similarly, this behaviour can be visually seen by comparing the original sequence in Fig 5d to the modified sequence in Fig 5e. Besides, the objects in the frames of the Football1 sequence in Fig 6b are almost captured by all of the eight trackers, while the proposed modified Football1 sequence (i.e., Football1_Modf), in Fig 6c, the performance of the trackers slightly degraded.

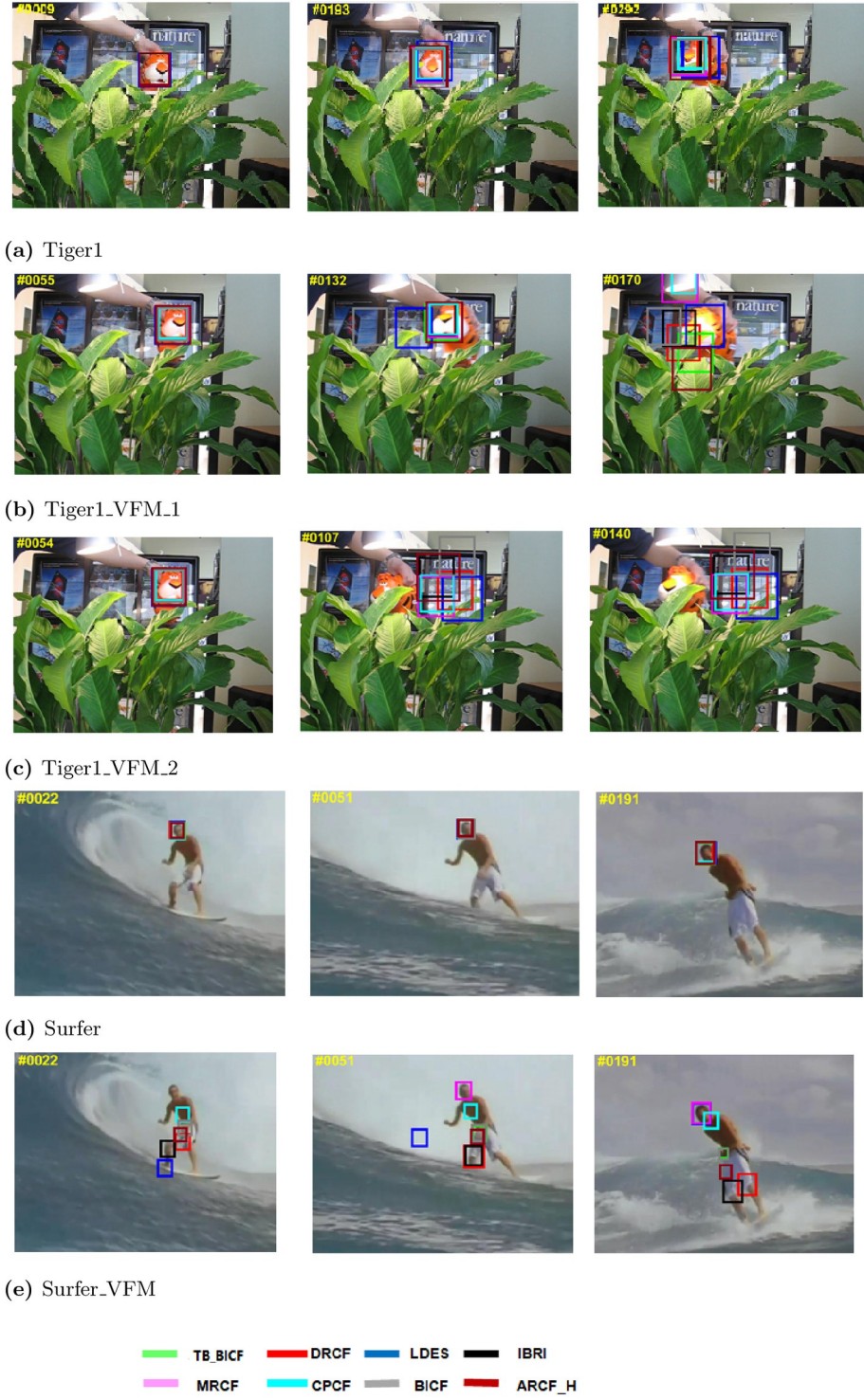

**Fig 5. Qualitative assessment of the tested trackers on the a) Tiger1, b) Tiger1_VFM_1, c) Tiger1_VFM_2, d) Surfer, and e) Surfer_VFM sequences [8].**

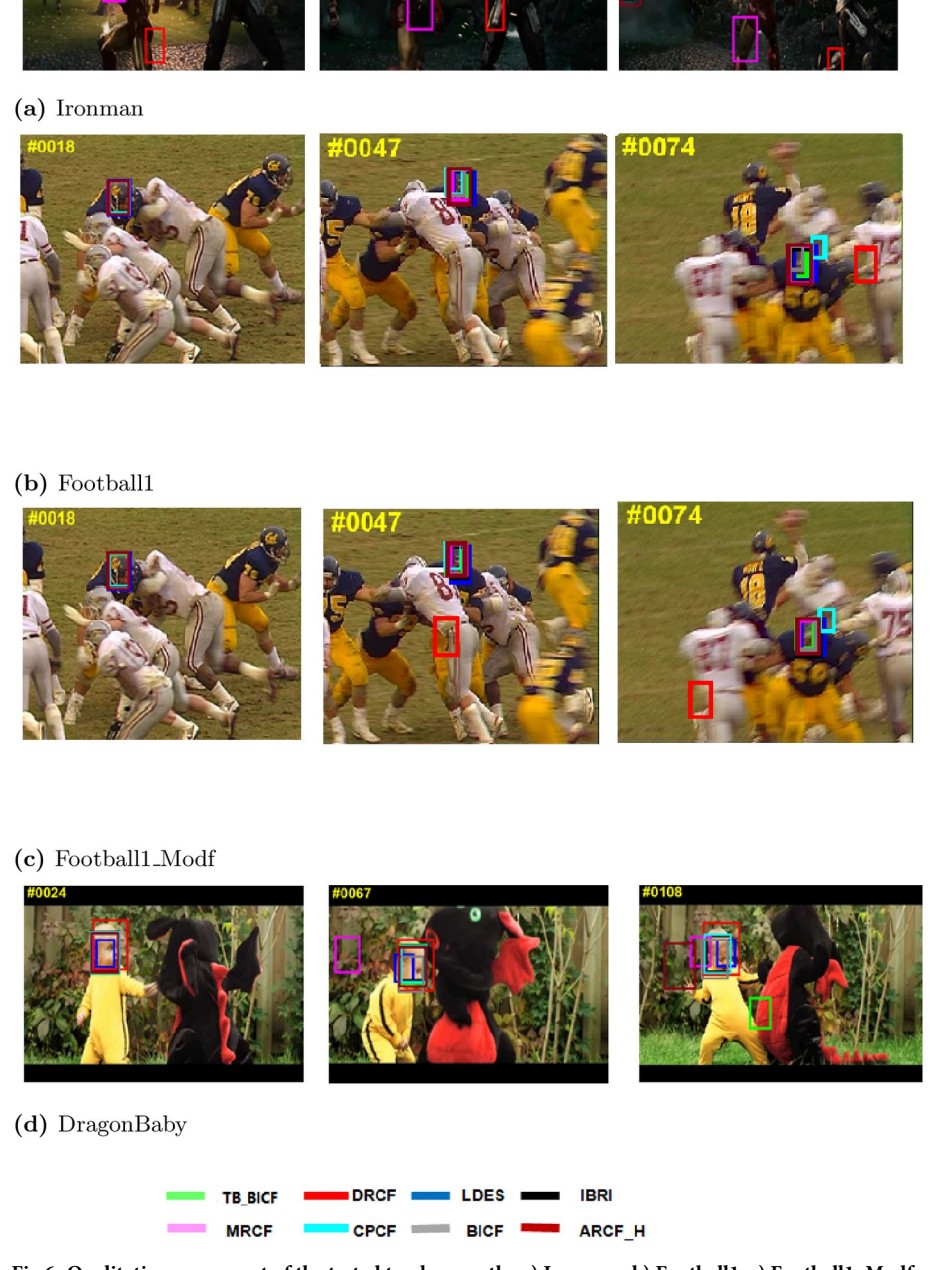

**(a)** Ironman

**(b)** Football1

**(c)** Football1_Modf

**(d)** DragonBaby

| | | | |
|---|---|---|---|
| TB_BICF | DRCF | LDES | IBRI |
| MRCF | CPCF | BICF | ARCF_H |

**Fig 6. Qualitative assessment of the tested trackers on the a) Ironman, b) Football1, c) Football1_Modf and d) DragonBaby video sequences [8].**

Second, we proposed using three metrics for the quantitative evaluation. Using the first metric, the performance of the eight trackers are evaluated in terms of the throughput, i.e., speed of the trackers to process an input video sequence. A number of factors influence the computational performance of CF-based trackers. These factors include the target object's bounding box size, the bounding box for searching area, the number of iterations, and the

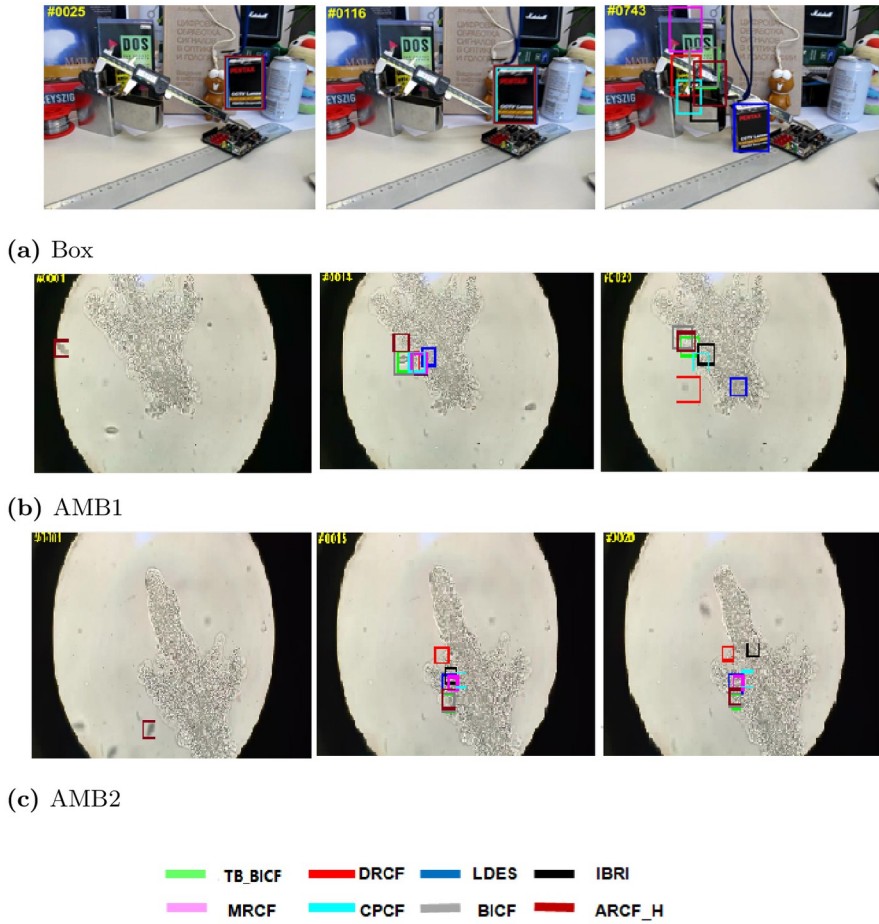

(a) Box

(b) AMB1

(c) AMB2

| ▬ TB_BICF | ▬ DRCF | ▬ LDES | ▬ IBRI |
|---|---|---|---|
| ▬ MRCF | ▬ CPCF | ▬ BICF | ▬ ARCF_H |

**Fig 7. Qualitative assessment of the tested trackers on the a) Box [8], b) AMB1 and c) AMB2 sequences.**

number of features per sequence. Table 2 lists the the eight trackers' average frame per second results for each tracker where the running times can be classified into three groups, namely, 1) slow, 2) fast, and 3) normal. The slow group of the largest running times trackers includes LDES and IBRI. The slowest performance of the LDES tracker can be linked to the idea, which is proposed by LDES, of using two boundary boxes to track the object of interest. This proposed idea of using two boundary boxes increases the running time drastically. The slow running time of IBRI tracker can be linked to the extended use of historical frames to track the object. The main motivation of IBRI tracker was that most of the existing methods combine the information of only two successive video frames; thus, IBRI extended the number of used frames at the cost of the running time. The fast group of the trackers includes the CPCF tracker only with almost 12 frames per second. The fast performance the CPCF tracker can be justified by utilizing two fast to compute two features, HoG and CN; computing those two features

**Table 2. The average fps for the eight trackers.**

| Visual Tracking Methods | BICF | TB_BICF | CPCF | DRCF | IBRI | ARCF_H | LDES | MRCF |
|---|---|---|---|---|---|---|---|---|
| fps | 8.48 | 9.93 | 12.18 | 9.65 | 6.27 | 10.25 | 3.41 | 10.46 |

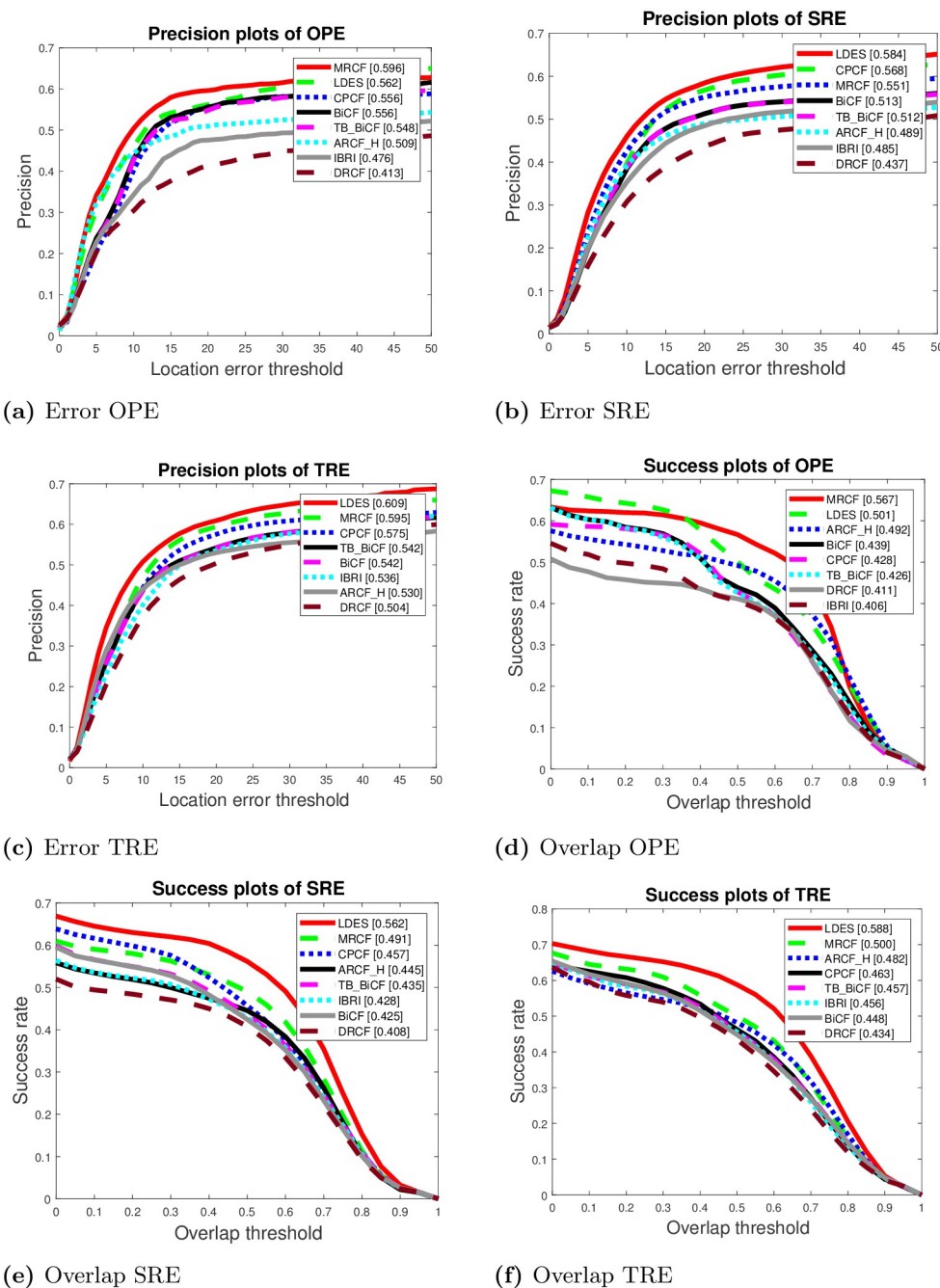

**Fig 8. Success and precision plots of error and overlap for OPE, SRE, and TRE.**

includes low computational requirements. The third group includes the remaining other six trackers which are close to each other in terms of the fps metric.

The second quantitative metric includes precision and success plots of OPE, TRE, and SRE. Fig 8 illustrates the precision and success plots of OPE, TRE, and SRE metrics for the eight trackers. Obviously, the LDES tracker has the best precision and success rates for the three metrics except for OPE success rate, as the best tracker was MRCF. Thus, the overall results show that superior results of LDES followed by MRCF while the worst performance yielded by DRCF tracker. Besides, Fig 9 depicts the success plots for 11 different attributes of the utilized

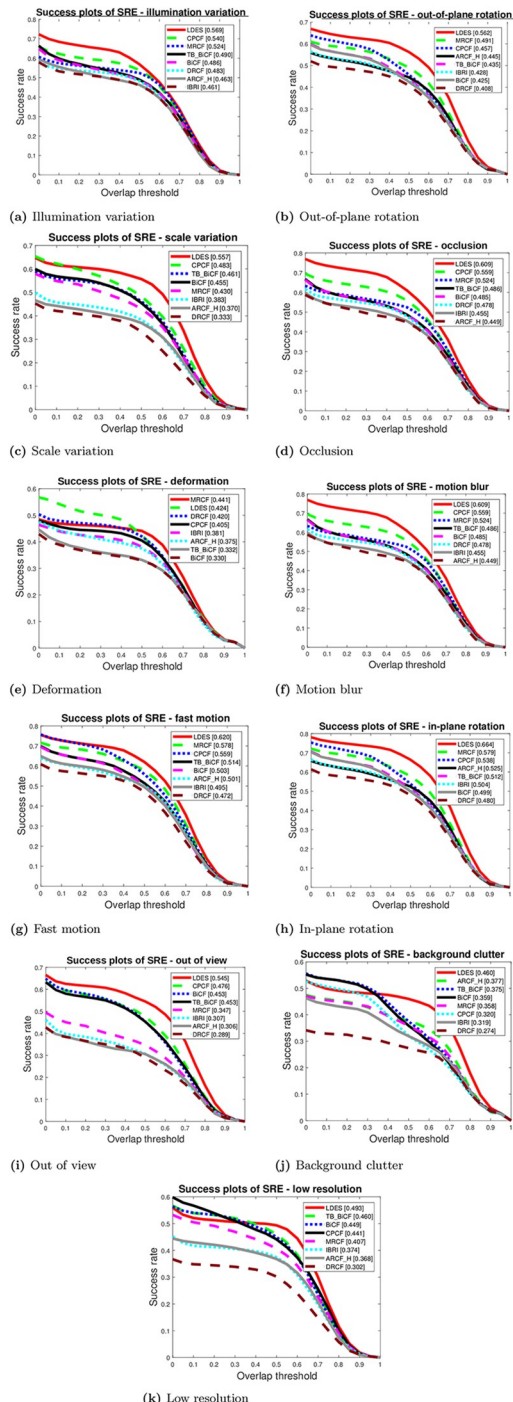

**Fig 9. Attribute-based success plots of SRE (illumination variation, out-of-plane rotation, scale variation, occlusion, deformation, and motion blur).** (a) Illumination variation. (b) Out-of-plane rotation. (c) Scale variation. (d) Occlusion. (e) Deformation. (f) Motion blur. (g) Fast motion. (h) In-plane rotation. (i) Out of view. (j) Background clutter. (j) Background clutter.

**Table 3. The percentage of frames the trackers managed to capture the object from the entire sequence.**

|  | ARCF_H | BICF | TB_BICF | CPCF | LDES | MRCF | DRCF | IBRI |
|---|---|---|---|---|---|---|---|---|
| Tiger1(354) | 346(97.7%) | 346(97.7%) | 346(97.7%) | 354(100%) | 330(93.2%) | 354(100%) | 354(100%) | 354(100%) |
| Tiger1_VFM_1(294) | 281(95.6%) | 84(28.6%) | 124(42.2%) | 125(42.5%) | 78(26.5%) | 125(42.5%) | 277(94.2%) | 275(93.5%) |
| Tiger1_VFM_2(324) | 256(79.8%) | 64(19.8%) | 64(19.8%) | 278(85.8%) | 64(19.8%) | 280(86.4%) | 65(20%) | 276(85.2%) |
| Surfer(376) | 376(100%) | 376(100%) | 376(100%) | 376(100%) | 376(100%) | 376(100%) | 376(100%) | 376(100%) |
| Surfer_VFM(366) | 15(4.1%) | 15(4.1%) | 15(4.1%) | 15(4.1%) | 306(83.6%) | 355(97%) | 15(4.1%) | 15(4.1%) |
| Football1(74) | 74(100%) | 74(100%) | 74(100%) | 73(98.6%) | 74(100%) | 74(100%) | 55(74.3%) | 74(100%) |
| Football1_Modf(67) | 74(100%) | 74(100%) | 74(100%) | 73(98.6%) | 74(100%) | 74(100%) | 23(34.3%) | 74(100%) |
| AMB1(20) | 6(30%) | 18(90%) | 18(90%) | 13(65%) | 12(60%) | 12(60%) | 16(80%) | 13(65%) |
| AMB2(20) | 4(20%) | 4(20%) | 4(20%) | 5(25%) | 4(20%) | 4(20%) | 4(20%) | 4(20%) |

datasets. In ten of these attributes, the LDES tracker again has the best performance followed by MRCF and ARCF_H while the only attribute the LDES has the second place was the deformation attribute.

The third quantitative metric is the summary of the number of frames captured by the trackers for the video sequences listed in Table 1. Table 3 lists the percentage of successfully captured framed. The overall results of Table 3 show that the eight trackers managed to track the objects in almost all of the frames of the original OTB-2013 video sequences, namely, Tiger1, Surfer, and Football1. This can be linked to the fact that the performance of the trackers are tuned according to the standard datasets. On the contrary, all of the trackers failed to track the objects of the proposed modified sequences in all of the frames of the modified sequences when the number of dropped frames are large, five or more frames.

The results of Table 3 outline the degrade in the eight trackers' performance when comparing the original video sequence of Tiger1 and Surfer at one side with the modified sequences Tiger1_VFM_1, Tiger1_VFM_2, and Surfer_VFM at the other side. For instance, the LDES, the best tracker of the previously discussed metrics, managed to capture 93.2% of the original OTB-2013 Tiger1 video sequence frames, which contains 354 frames. This performances, of LDES tracker, degraded to 26.5% and 19.8% for the proposed modified sequences of Tiger1_VFM_1 and Tiger1_VFM_2, respectively. The LDES tracker managed to capture the object in 330 frames out of 354 frames of the original Tiger1 video sequence while the LDES tracker managed to captured the object only in 78 frames out of 294 frames of the proposed modified video sequence Tiger1_VFM_1. The performance of the eight trackers degraded on the modified sequences but with varies ratios based on the tracker parameters, e.g., bounding box size, for searching area in the frame, the number of considered previous frames. Of note, the same results can be found for the Surfer sequence and its modified sequence as well. To study the effect of the number of dropped frames on the performance deterioration, we modified the Tiger1, Surfer, and Football1 video sequence of OTB-2013 by dropping 10, 5, and 1 consecutive frames, respectively. As the Football1 is modified by dropping one frame each ten frames, the trackers' performance is almost fixed, almost no effect is noticed. Thus, the recommended number of dropped frames should be five or more consecutive frames, as the trackers' performance deteriorated for the modified sequences of Surfer and Tiger1 only. Finally, the intervals of the missing frames of each sequences for each tracker are listed in Table 4, where N/A means the tracker did not miss the object in any frame.

In Table 3, the results of AMB1 and AMB2 video sequence are consistence with the motivation of the proposed feature, i.e., *hard-to-follow-by-human*, as the more difficult to follow the object by human's naked eyes, the worst the performance of the tracker. The AMB2 video

**Table 4. The frame ranges of CF-based trackers when failing to capture objects.**

|  | ARCF_H | BICF | TB_BICF | CPCF | LDES | MRCF | DRCF | IBRI |
|---|---|---|---|---|---|---|---|---|
| Tiger1(354) | (347–354) | (347–354) | (347–354) | N\A | (60–64) (67–71) (94–110) | N\A | N\A | N\A |
| Tiger1_VFM_1(294) | (51–56) (131–139) | (51–57) (91–294) | (51–57) (131–294) | (51–56) (131–294) | (51–57) (85–294) | (51–56) (131–294) | (51–56) (131–142) | (51–58) (131–142) |
| Tiger1_VFM_2(324) | (51–56) (71–125) (316–324) | (51–57) (71–324) | (51–57) (71–324) | (51–56) (71–95) (108–125) | (51–57) (71–324) | (51–56) (71–94) (108–124) | (51–56) (71–324) | (51–59) (71–94) (108–125) |
| Surfer(376) | N\A | N\A | N\A | N\A | N\A | N\A | N\A | N\A |
| Surfer_VFM(366) | (16–366) | (16–366) | (16–366) | (16–366) | (16–76) | (16–26) (366) | (16–366) | (16–366) |
| Football1(74) | N\A | N\A | N\A | (74) | N\A | N\A | (26–28) (58–74) | N\A |
| Football1_Modf(67) | N\A | N\A | N\A | (74) | N\A | N\A | (24–67) | N\A |
| AMB1(20) | (7–20) | (18,19) | (19,20) | (14–20) | (13–20) | (13–20) | (17–20) | (14–20) |
| AMB2(20) | (5–20) | (5–20) | (5–20) | (6–20) | (5–20) | (5–20) | (4) (6–20) | (5–20) |

sequence is more difficult to follow for human's eyes than AMB1; this finding is emphasized by results of Table 3.

The results of Figs 8 and 9 are the standard curves to evaluate the eight trackers. These two figures show that the LDES tracker has the overall best performance. On the contrary, with a deeper analysis of the results in Tables 3 and 4 of the proposed modified video sequences, the results reveal that the LDES tracker is not the best tracker. For instance, Table 3 shows that the best trackers of the proposed modified sequences Tiger1_VFM_1 and Tiger1_VFM_1 are ARCF_H and MRCF, respectively. The LDES tracker was the worst tracker in terms of object capturing on the proposed modified sequence Tiger1_VFM_1; this is exactly the opposite to the results of the standard dataset of Figs 8 and 9. Despite the average performance of the BICF and CPCF trackers relative to the LDES tracker's performance in Figs 8 and 9, the best trackers were BICF and CPCF for the proposed AMB1 and AMB2 video sequences, respectively. For the proposed video sequences, the LDES tracker performance was average in comparison to the other trackers. These results emphasize that there is a need to test the CF-based trackers on video sequences rather than the sequences of the standard datasets.

## 6 Conclusion

The CF-based trackers are the most suitable visual object tracker type for real-time applications such as AUV. In this paper, we proposed to provide a practical evaluation of the most recent well-performing CF-based trackers. The lack of such practical evaluations motivated this work. In this context, we carefully selected eight CF-based trackers based on their performance on the state-of-the-art datasets (e.g., OTB-2013 and OTB-2015). Those eight trackers are theoretically discussed in terms of their merit and demerit. Then, we propose to practically evaluate those trackers by utilizing the OTB-2013 dataset and proposing a set of new video sequences with new features which are not been included in the standard object tracking datasets before, namely, *hard-to-follow-by-human* and *very fast motion*. The proposed feature *hard-to-follow-by-human* merges both the background clutters and occlusions features in a unique manner. As it can be concluded from the new feature name, the video sequence includes an object which is difficult, but not impossible, for a human to track. In addition, we

proposed modifying a number of existing video sequences of OTB-2013 with the fast motion feature; the modification includes increasing the speed of motion by dropping more frames with different scenarios. Considering the OTB-2103 dataset and the new video sequences, the eight trackers are thoroughly evaluated on different metrics, including precision and success rate for quantitative assessment. Finally, all of the eight CF-trackers failed to capture the object in every frame in the proposed sequences while the same trackers managed to capture the object in almost every frame of the sequences of the standard dataset. The performance of the eight trackers was very poor on the video sequence with the proposed features *hard-to-follow-by-human*. The future directions include improving the CF-based filter to manage to better capture objects in sequences with the proposed *hard-to-follow-by-human* feature and comparing other object tracking datasets, as this study was limited only to the OTB-2013 dataset.

## Supporting information

**S1 File.**
(ZIP)

## Author Contributions

**Conceptualization:** Ahmad Salah.

**Data curation:** Ahmed W. Sallam, Andrew Gatt.

**Investigation:** Andrew Gatt.

**Methodology:** Islam Mohamed, Andrew Gatt, Ahmad Salah.

**Project administration:** Ahmad Salah.

**Software:** Islam Mohamed.

**Supervision:** Ibrahim Elhenawy, Ahmad Salah.

**Validation:** Ibrahim Elhenawy, Ahmad Salah.

**Visualization:** Islam Mohamed, Ibrahim Elhenawy, Ahmed W. Sallam.

**Writing – original draft:** Islam Mohamed, Ibrahim Elhenawy, Ahmed W. Sallam.

**Writing – review & editing:** Islam Mohamed, Ibrahim Elhenawy, Ahmed W. Sallam, Ahmad Salah.

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
