## [Decision Letter · Decision Letter 0]

15 Mar 2022

PONE-D-21-40546A practical evaluation of correlation filter-based object trackers with new datasetsPLOS ONE

Dear Dr. hassan,

Thank you for submitting your manuscript to PLOS ONE. After careful consideration, we feel that it has merit but does not fully meet PLOS ONE’s publication criteria as it currently stands. Therefore, we invite you to submit a revised version of the manuscript that addresses the points raised during the review process.

We look forward to receiving your revised manuscript.

Kind regards,

Mohamed Hammad, Ph.D.

Academic Editor

PLOS ONE

Journal Requirements:

2. PLOS requires an ORCID iD for the corresponding author in Editorial Manager on papers submitted after December 6th, 2016. Please ensure that you have an ORCID iD and that it is validated in Editorial Manager. To do this, go to ‘Update my Information’ (in the upper left-hand corner of the main menu), and click on the Fetch/Validate link next to the ORCID field. This will take you to the ORCID site and allow you to create a new iD or authenticate a pre-existing iD in Editorial Manager. Please see the following video for instructions on linking an ORCID iD to your Editorial Manager account: https://www.youtube.com/watch?v=_xcclfuvtxQ.

3. We note that Figures 4, 6, 7 in your submission contain copyrighted images. All PLOS content is published under the Creative Commons Attribution License (CC BY 4.0), which means that the manuscript, images, and Supporting Information files will be freely available online, and any third party is permitted to access, download, copy, distribute, and use these materials in any way, even commercially, with proper attribution. For more information, see our copyright guidelines: http://journals.plos.org/plosone/s/licenses-and-copyright.

   1. You may seek permission from the original copyright holder of Figures 4, 6, 7to publish the content specifically under the CC BY 4.0 license.

Reviewers' comments:

Reviewer's Responses to Questions

**Comments to the Author**

1. Is the manuscript technically sound, and do the data support the conclusions?

Reviewer #1: Yes

Reviewer #2: Partly

2. Has the statistical analysis been performed appropriately and rigorously? 

Reviewer #1: Yes

Reviewer #2: Yes

3. Have the authors made all data underlying the findings in their manuscript fully available?

Reviewer #1: Yes

Reviewer #2: Yes

4. Is the manuscript presented in an intelligible fashion and written in standard English?

Reviewer #1: No

Reviewer #2: Yes

5. Review Comments to the Author

Reviewer #1: There are clear weaknesses in the paper that the authors must particularly pay attention and handle:

• The major problem of this work is that its novelty is very limited where some methods have already been used in the literature. In addition, the theoretical contribution is so limited. So, the authors should modify it carefully and improve the novelty of this paper. Also, the authors should provide solid motivation for their work based on the existing literature.

• The first occurrence of abbreviations of technical terms should be explained with the full name. For example, "LDES" in Abstract.”

• Figures. 9 and 10 need to be amended, where the font in figures is slightly small to be seen which make them difficult to read.

• The review for most of the cited work is nebulous. The review failed to summarize critical details about each of the cited research study and its relevant pros and cons.

• Section 3 and 4 need more explanations and details which are the core of this paper.

• The results should be further analyzed, more details and further discussion of the simulation results is needed.

• The conclusions section should conclude that you have achieved from the study, contributions of the study to academics and practices. In addition, list the advantages and disadvantages of the proposed solution, as well as indicate the limitations of work. Further, mention the recommendations of future works.

• The writing of the paper needs work. It is full of syntactical and grammatical mistakes, that make its understandability extremely difficult. A thorough proofreading is required (best by a native English speaker).

• The list of references should be reformatted and checked again to be matched with the journal requirement where a different styles and types are used. Please check some spells and typos. In addition, I recommend the authors to read and cite the following paper which is recently done and will be helpful to the revision of this paper:

1. " A background-aware correlation filter with adaptive saliency-aware regularization for visual tracking.", 2022.

2. " Liu, Bing, et al. "HCDC-SRCF tracker: Learning an adaptively multi-feature fuse tracker in spatial regularized

correlation filters framework.", 2022.

3. "Accelerated duality-aware correlation filters for visual tracking." Neural Computing and Applications.” 2022.

4. " Visual tracking: Tracking in scenes containing multiple moving objects.” 2022.

Reviewer #2: The paper proposed a set of new sequences with higher difficulties than the existing object tracking benchmarking sequences, and evaluated eight CF-based trackers through a set of experiments.

The paper introduced basic concepts such as simple correlation and general correlation filter-based trackers very well, however, recent research (papers in recent one or two years) may be overlooked.

The three main contributions claimed in Introduction were not supported sufficiently by the following sections.

In addition, here is some other advice.

1. It is required to give the full form of abbreviations when they first appear. Eg., abbreviations on P2.

2. What does it mean by "The two images in Fig. 3…" in the last paragraph on P3?

3. Differences between OTB original sequences and the modified sequences can not be shown clearly on Fig. 4. It would be better to present another better expression method.

4. What are main differences between microscopic video sequences and macroscopic video sequences? Since it was claimed to be proposed for the first time, it would be better to pay more attention to depict and discuss about these microscopic sequences.

5. It may be regarded as a main contribution to propose a set of new sequences, then it would be better to explain the idea and processes more detailed, such as why to choose these criteria, how to highlight the innovation, etc. Especially, how significant are differences between OTB original sequences and the modified sequences? How to measure the significance of differences?

6. After describing expriment results, it is supposed to present some deep discussion. Why does one tracker outperform others? Otherwise, the contribution is not persuasive

7. It would be better to further refine the innovation and contribution.

6. PLOS authors have the option to publish the peer review history of their article (what does this mean?). If published, this will include your full peer review and any attached files.

Reviewer #1: No

Reviewer #2: No

---

## [Author Response · Author response to Decision Letter 0]

11 Jul 2022

Kindly find the response letter enclosed

---

## [Decision Letter · Decision Letter 1]

2 Aug 2022

A practical evaluation of correlation filter-based object trackers with new features

PONE-D-21-40546R1

Dear Dr. Moustafa,

We’re pleased to inform you that your manuscript has been judged scientifically suitable for publication and will be formally accepted for publication once it meets all outstanding technical requirements.

Kind regards,

Mohamed Hammad, Ph.D.

Academic Editor

PLOS ONE

Additional Editor Comments (optional):

Reviewers' comments:

Reviewer's Responses to Questions

**Comments to the Author**

1. If the authors have adequately addressed your comments raised in a previous round of review and you feel that this manuscript is now acceptable for publication, you may indicate that here to bypass the “Comments to the Author” section, enter your conflict of interest statement in the “Confidential to Editor” section, and submit your "Accept" recommendation.

Reviewer #1: All comments have been addressed

Reviewer #3: All comments have been addressed

2. Is the manuscript technically sound, and do the data support the conclusions?

Reviewer #1: Yes

Reviewer #3: Yes

3. Has the statistical analysis been performed appropriately and rigorously? 

Reviewer #1: Yes

Reviewer #3: Yes

4. Have the authors made all data underlying the findings in their manuscript fully available?

Reviewer #1: Yes

Reviewer #3: Yes

5. Is the manuscript presented in an intelligible fashion and written in standard English?

Reviewer #1: Yes

Reviewer #3: Yes

6. Review Comments to the Author

Reviewer #1: Since the previous version, authors have done huge work and the paper is much better. This version looks good Therefore, I suggest accepting this paper in the current form.

Reviewer #3: In this paper, The authors provide a practical evaluation of the most recent well-performing CF-based trackers. The lack of such practical evaluations motivated this work. In this context, they carefully select eight CF-based trackers based on their performance on the state-of-the-art datasets (e.g., OTB-2013 and OTB-2015). Those eight trackers are theoretically discussed in terms of their merit and demerit. Then, they propose to practically evaluate those trackers by utilizing the OTB-2013 dataset and proposing a set of new video sequences with new features which are not been included in the standard object tracking datasets before, namely, hard-to-follow-by-human and very fast motion.

This version of the manuscript is well improved and the results seem correct. I recommend this manuscript for publication in the present form.

7. PLOS authors have the option to publish the peer review history of their article (what does this mean?). If published, this will include your full peer review and any attached files.

Reviewer #1: No

Reviewer #3: No

---

## [Editor Report · Acceptance letter]

5 Aug 2022

PONE-D-21-40546R1 

A practical evaluation of correlation filter-based object trackers with new features 

Dear Dr. Salah:

I'm pleased to inform you that your manuscript has been deemed suitable for publication in PLOS ONE. Congratulations! Your manuscript is now with our production department. 

Kind regards, 

on behalf of

Dr. Mohamed Hammad 

Academic Editor

PLOS ONE